# UniHDA: A Unified and Versatile Framework for Generalized Hybrid Domain Adaptation

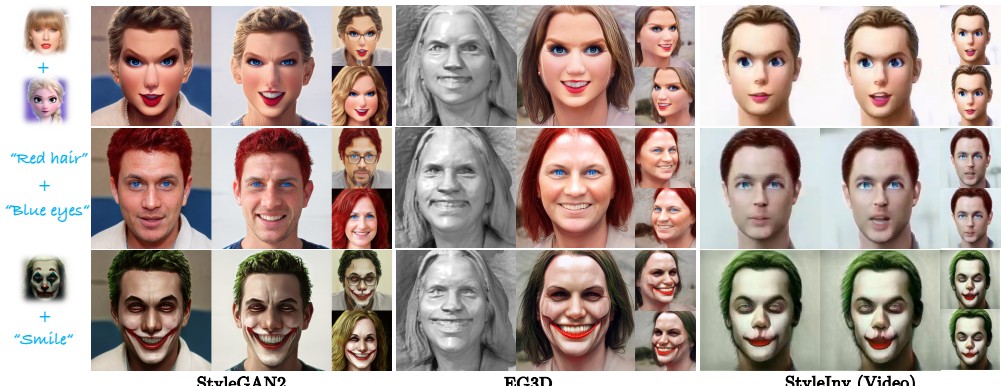

Figure 1: Given a pre-trained generator and multiple domains, UniHDA adapts the generator to a hybrid target domain that blends all characteristics at once and maintains robust cross-domain consistency. It supports both image and text modalities and is versatile to multiple generators.

## Abstract

Recently, generative domain adaptation has achieved remarkable progress, enabling us to adapt a pre-trained generator to a new target domain. However, existing methods are limited to a single target domain and single modality, either text-driven or image-driven. In this paper, we explore a novel task – *Generalized Hybrid Domain Adaptation*. Compared with conventional generative domain adaptation, it provides greater flexibility to adapt the generator to the hybrid of multiple target domains, with multi-modal references including one-shot image and zero-shot text prompt. Meanwhile, it is more challenging to represent the composition of multi-modal target domains and preserve the characteristics from the source domain. To address these issues, we propose UniHDA, a **unified** and **versatile** framework for generalized hybrid domain adaptation. Drawing inspiration from the interpolable latent space of StyleGAN, we find that a linear interpolation between domain shifts in CLIP's embedding space can also uncover favorable compositional capabilities for the adaptation. In light of this finding, we linearly interpolate the domain shifts from multiple target domains to achieve hybrid domain adaptation. To enhance **consistency** with the source domain, we further propose a novel cross-domain spatial structure (CSS) loss that maintains the detailed spatial structure between the source and target generator. Experiments show the adapted generator can synthesize realistic images with various attribute compositions and maintain robust consistency with the source domain. Additionally, UniHDA is generator-agnostic and versatile to multiple generators, *e.g.*, StyleGAN, EG3D, and video generators.

## 1 Introduction

Benefiting from the tremendous success of modern image generators (Karras et al., 2019; Brock et al., 2018; Vahdat et al., 2021; Rombach et al., 2022), generative domain adaptation has achieved remarkable progress. Typically, it aims to adapt a pre-trained generator to a new target domain while preserving the variation in the source domain. Depending on the modality of references, generative

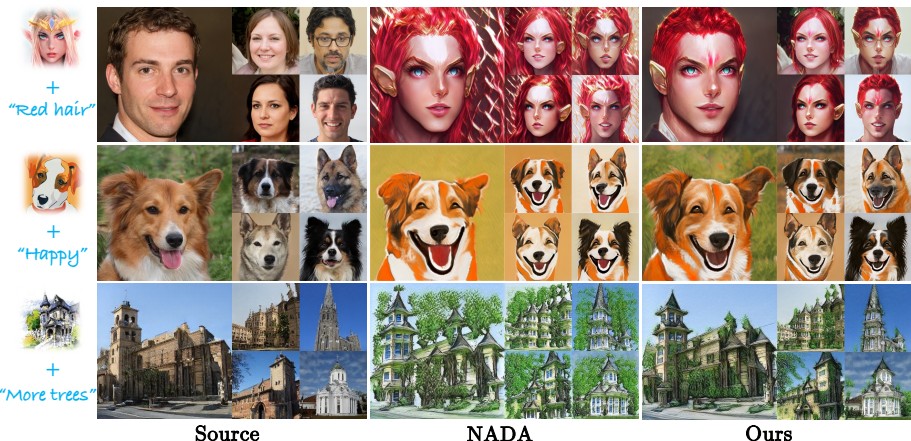

Source     NADA     Ours

Figure 2: UniHDA succeeds in generalized hybrid domain adaptation and maintains robust consistency with the source domain. NADA (Gal et al., 2021), by cross-model interpolation, can somehow yield alike images of hybrid domains but with poor consistency. It overfits the limited references, which impedes the inheritance of the diversity in the source domain.

domain adaptation can be categorized into two schools: text-driven (Gal et al., 2021; Nitzan et al., 2023a; Liu et al., 2023; Song et al., 2022; Lei et al., 2023) and image-driven (Li et al., 2020; Ojha et al., 2021; Zhao et al., 2022b; Xiao et al., 2022; Mondal et al.; Wu et al., 2023; Zhao et al., 2022a).

Despite their promising results for single modality and single target domain, they fail to adapt the generator to more practical scenarios like `smiling Joker` given the non-smiling `Joker` image and the `smile` text. For more general purposes, we explore a new task – *Generalized Hybrid Domain Adaptation*. It aims to adapt the generator to the hybrid of multiple target domains that integrates all attributes with multi-modal references including one-shot image and zero-shot text prompt (Fig. 1).

Compared with conventional generative domain adaptation, our task is more challenging in two aspects: (1) It is harder to represent the composition of multi-modal target domains. While cross-model interpolation technique (Gal et al., 2021) could somehow yield alike images of the hybrid domain, it doubles the model size and training time to train a separate model per domain. (2) With multiple target domains and very limited references from each domain, the generator is more prone to overfitting domain-specific attributes. This leads to a loss of consistency with the source domain, which impedes the inheritance of the diversity (Fig. 2).

To address these issues, we propose UniHDA, a **Uni**fied and versatile framework for Generalized **H**ybrid **D**omain **A**daptation. UniHDA facilitates the references of one-shot image and text prompt simultaneously and blends the attributes from target domains to create a hybrid domain. To enable multiple modalities, we leverage pre-trained CLIP (Radford et al., 2021) to project multi-modal references into a unified embedding space and represent the domain shift by the direction vector from the source embedding to the target embeddings.

To achieve hybrid domain adaptation, we draw inspiration from the compositional capabilities in the latent space of StyleGAN (Härkönen et al., 2020; Shen & Zhou, 2021; Xu et al., 2022). Specifically, we demonstrate a semantically meaningful linear interpolation between direction vectors in CLIP's embedding space can uncover favorable compositional capabilities (Fig. 3). In light of this intriguing finding, we linearly interpolate direction vectors of multiple target domains to obtain the direction vector corresponding to the hybrid domain that semantically integrates attributes from all target domains.

Furthermore, we introduce a novel Cross-domain Spatial Structure loss (CSS) to preserve the consistency between the source and target generator by maintaining detailed spatial structure information. Concretely, we leverage pre-trained Dino-ViT (Dosovitskiy et al., 2020; Oquab et al., 2023) to encode generated images into patch tokens with fine-grained spatial information. For cross-domain consistency, we maintain the correspondence between source and target tokens with contrastive learning (Oord et al., 2018). Equipped with CSS loss, UniHDA maintains robust consistency with the source domain as shown in Fig. 2.

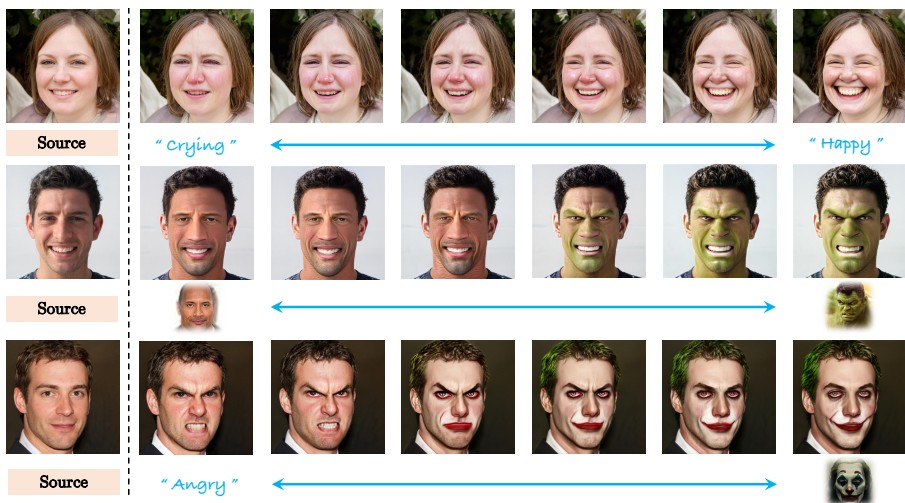

Figure 3: In CLIP's embedding space, we find direction vectors from source domain to target domains are linearly interpolable to represent the domain shift to the hybrid target domain. As shown above, the traversal portrays smooth transition between two target domains and shows favorable compositional capabilities to semantically integrates the attributes. Note that the coefficients for the right domain are respectively 0, 0.2, 0.4, 0.6, 0.8, and 1, while they are inverse for the left domain.

We conduct experiments for a wide range of source and target domains to validate the effectiveness of our method. Results demonstrate that the adapted generator can synthesize realistic images with various attribute compositions. We also show that UniHDA is agnostic to the type of generators, *e.g.*, StyleGAN (Karras et al., 2019; 2020; 2021), EG3D (Chan et al., 2022), video model (Wang et al., 2023), and Diffusion models (Ho et al., 2020; Kim et al., 2022a). Our contributions are as follows:

- We explore a novel task *Generalized Hybrid Domain Adaptation* that offers greater flexibility for hybrid target domain and multi-modal references. To enable it, we propose a **unified** and **versatile** framework which effectively accomplishes the adaptation and is versatile for various generators.
- We demonstrate strong compositional capabilities of direction vectors in CLIP's embedding space. Taking advantage of it, we propose to linearly interpolate the direction vectors for generalized hybrid domain adaptation.
- We propose a cross-domain spatial structure loss to maintain consistency with the source domain. It is conducted in generator-agnostic embedding space which is versatile for various generators. To our knowledge, it is the very first trial in generative domain adaptation.

## 2 RELATED WORK

**Text-driven Generative Domain Adaptation.** Text-driven domain adaptation (Gal et al., 2021; Nitzan et al., 2023a; Liu et al., 2023; Alanov et al., 2022; Zhu et al., 2022; Lyu et al., 2023; Lei et al., 2023; Kim & Chun, 2023; Kim et al., 2023; Song et al., 2022) involves using a textual prompt to shift the domain of a pre-trained model toward a new domain. For example, Style-NADA (Gal et al., 2021) presents a local direction CLIP (Radford et al., 2021) loss to align the embeddings of the generated images and text. Based on Style-NADA, Domain Expansion (DE) (Nitzan et al., 2023b) proposes to expand the generator to jointly model multiple domains with texts.

**Image-driven Generative Domain Adaptation.** Image-driven generative domain adaptation (Mo et al., 2020; Li et al., 2020; Ojha et al., 2021; Zhao et al., 2022b; Xiao et al., 2022; Mondal et al.; Wu et al., 2023; Zhao et al., 2022a; 2023; Zhang et al., 2022; Zhu et al., 2021; Alanov et al., 2023; Kim et al., 2022b) refers to the adaptation of a pre-trained image generator to a new target domain using a limited number of training images. Prior methods often integrate additional regularization terms to prevent overfitting. For instance, DiFa (Zhang et al., 2022) utilizes GAN inversion (Tov et al., 2021) to align the latent codes which helps inherit diversity. Although these works have made significant strides in generative domain adaptation, they heavily rely on the discriminator or generator, making it challenging to handle hybrid domain adaptation and extend to other generators.

**Generative Hybrid Domain Adaptation.** To achieve hybrid domain adaptation, Style-NADA (Gal et al., 2021) proposes to train a separate generative model per domain and combine their effects in test-time. However, it doubles the model size and training time. Domain Expansion (DE) (Nitzan et al., 2023b) expands the generator to jointly model multiple domains via decomposing latent space. However, it requires the source dataset for regularization, which significantly increases training time. Recently, FHDA (Li et al., 2023) proposes few-shot hybrid domain adaptation and introduces a directional subspace loss. Differently, we focus on multi-modal references including one-shot image, which offers greater flexibility.

**Disentanglement in Generative Models.** As observed in StyleGAN (Karras et al., 2019), the latent space is essentially a linear subspace. Recent works (Härkönen et al., 2020; Shen & Zhou, 2021; Xu et al., 2022; Shen et al., 2020; Wu et al., 2020; Patashnik et al., 2021; Voynov & Babenko, 2020; Spingarn-Eliezer et al., 2020) propose to find individual latent factors for image variations. Among them, SeFa (Shen & Zhou, 2021) computes the eigenvalues of the transformation matrix to find the latent directions. For diffusion models, DiffAE (Preechakul et al., 2022) explores the possibility of using DPMs for representation learning and extracts a decodable representation of an input image.

# 3 METHOD

## 3.1 GENERALIZED HYBRID DOMAIN ADAPTATION

We start with a pre-trained generator $G_{\mathcal{S}}$ (*e.g.*, StyleGAN (Karras et al., 2019; 2020; 2021) and Diffusion model (Ho et al., 2020; Song et al., 2020)), that maps from noise $z$ to images in a source domain $\mathcal{S}$. Given a new target domain $\mathcal{T}$ referenced by texts (Gal et al., 2021; Zhang et al., 2022; Kwon et al., 2022; Kim et al., 2022a) or images (Mo et al., 2020; Li et al., 2020; Ojha et al., 2021; Zhao et al., 2022b; Xiao et al., 2022; Mondal et al.), generative domain adaptation aims to adapt $G_{\mathcal{S}}$ to yield a target generator $G_{\mathcal{T}}$, which can generate images similar to domain $\mathcal{T}$.

Despite the promising results of existing methods, a major limitation of them is that they only support adaptation from the source domain to individual target domains and fail to directly adapt the generator to the hybrid domain which blends the characteristics of multiple domains. Furthermore, they fail with multi-modal adaptation driven by texts and images simultaneously.

For more general purposes, we explore generalized hybrid domain adaptation. Given $N$ domains $\{\mathcal{T}_i\}_{i=1}^N$ with one-shot image $\{Y_i\}$ and $M$ domains $\{\mathcal{T}_j\}_{j=1}^M$ with the text prompt $\{P_j\}$, it aims to adapt the source generator $G_{\mathcal{S}}$ to $G_{\mathcal{T}}$ that models the hybrid domain $\mathcal{T} = \{\mathcal{T}_i\} \cup \{\mathcal{T}_j\}$ and generates images with integrated characteristics. To the end, we introduce UniHDA, a unified and versatile framework for generalized hybrid domain adaptation (Fig. 4).

## 3.2 MULTI-MODAL DIRECTION LOSS

To enable multiple modalities, we leverage pre-trained CLIP (Radford et al., 2021) to encode text-image references into a unified semantic embedding space. Drawing inspiration from CLIP-based methods (Gal et al., 2021; Zhang et al., 2022; Kwon et al., 2022; Kim et al., 2022a), we represent the *domain shift* as the direction vector $\Delta f_{dom}$ from the source embedding to the target embedding. For image reference $Y_i$ and its CLIP embedding $f_i$, the *domain shift* is calculated by

$$\Delta f_{dom} = f_i - \overline{f_s}, \tag{1}$$

where $\overline{f_s}$ is the mean embedding of several samples generated by $G_{\mathcal{S}}$. For text prompt $P_j$ and its CLIP embedding $f_j$,

$$\Delta f_{dom} = f_j - \widetilde{f_s}, \tag{2}$$

where $\widetilde{f_s}$ is the embedding of the source text prompt.

To adapt $G_{\mathcal{S}}$, we initialize a new generator $G_{\mathcal{T}}$ from $G_{\mathcal{S}}$ and finetune it by aligning the *sample-shift* direction $\Delta f_{samp}$ with the *domain-shift* direction $\Delta f_{dom}$. Formally,

$$\Delta f_{samp} = f_t - f_s,$$
$$\mathcal{L}_{direct} = 1 - \frac{\Delta f_{samp} \cdot \Delta f_{dom}}{\|\Delta f_{samp}\| \, \|\Delta f_{dom}\|}, \tag{3}$$

where $f_s$ and $f_t$ are the embeddings of samples generated by $G_{\mathcal{S}}$ and $G_{\mathcal{T}}$ with the same noise.

Figure 4: Overview of UniHDA with multi-modal direction loss $\mathcal{L}_{direct}$ and cross-domain spatial structure loss $\mathcal{L}_{\text{CSS}}$. Utilizing CLIP image and text encoder, $\mathcal{L}_{direct}$ encourages $G_{\mathcal{T}}$ to faithfully acquire domain-specific characteristics with multi-modal references. To facilitate diversity inherited from $G_{\mathcal{S}}$, $\mathcal{L}_{\text{CSS}}$ improves cross-domain consistency by maintaining detailed spatial structure information. The red solid line represents positive pairs, while the red dashed lines represent negative pairs.

## 3.3 LINEAR COMPOSITION OF DIRECTION VECTORS

To achieve the hybrid domain adaptation, we draw inspiration from the compositional capabilities in the latent space of StyleGAN (Härkönen et al., 2020; Shen & Zhou, 2021; Xu et al., 2022). We illustrate that a linear interpolation between two direction vectors in the embedding space of CLIP, which is semantically meaningful, reveals promising compositional capabilities. As shown in Fig. 3, we can smoothly interpolate between two direction vectors calculated by distinct target prompts and source prompt "photo", resulting in a gradual adaptation toward the target domain.

In light of this intriguing finding, we employ linear interpolation on the direction vectors of multi-modal target domains, to derive the direction vector representing the hybrid domain that semantically integrates all attributes. For given domain coefficients $\{\alpha_i\}$ and $\{\alpha_j\}$, we obtain the direction vector

$$\Delta f_{dom} = \sum_{i=1}^{N} \alpha_i(f_i - \overline{f_s}) + \sum_{j=1}^{M} \alpha_j(f_j - \widetilde{f_s}), \tag{4}$$

which represents the *domain shift* between the hybrid domain and source domain. We then substitute Eq. (4) into Eq. (3) to adapt $G_{\mathcal{S}}$ to the hybrid domain.

## 3.4 CROSS-DOMAIN SPATIAL STRUCTURE LOSS

Albeit the direction loss achieves generalized hybrid domain adaptation, the adapted generator is prone to overfit domain-specific attributes. This exacerbates when it comes to image-image and image-text scenarios owing to the scarcity of the images. To address this issue, we introduce a novel cross-domain spatial structure loss (CSS) to enhance cross-domain consistency, ensuring the preservation of intricate spatial structural information between the source and target generator.

Specifically, we leverage pre-trained Dino-ViT (Dosovitskiy et al., 2020; Oquab et al., 2023) to encode the generated images into patch tokens, containing detailed spatial structural information. Dino-ViT is self-supervised to focus on the distinction between subjects of the same class (Ruiz et al., 2023), which facilitates us in maintaining cross-domain consistency. Motivated by contrastive learning (Oord et al., 2018), we reduce distances between the positive token pairs at the same position and push away the negative token pairs at other positions by

$$\mathcal{L}_{\text{CSS}} = -\sum_i \log \frac{\exp(v_i^t \cdot v_i^s)}{\sum_j \exp(v_i^t \cdot v_j^s)}, \tag{5}$$

where $v_i^t$ and $v_j^s$ are the $i$-th and $j$-th tokens in the last layer of Dino-ViT from $G_{\mathcal{T}}$ and $G_{\mathcal{S}}$ respectively. The dot mark $\cdot$ represents dot product.

Overall, our training loss consists of two terms, i.e., the multi-modal direction loss $\mathcal{L}_{direct}$ to achieve generalized hybrid domain adaptation and the cross-domain spatial structure loss $\mathcal{L}_{\text{CSS}}$ to maintain cross-domain consistency:

$$\mathcal{L}_{overall} = \mathcal{L}_{direct} + \lambda \mathcal{L}_{\text{CSS}}. \tag{6}$$

Figure 5: **Image-image** hybrid domain adaptation. We compare the results of FHDA (Li et al., 2023), NADA (Gal et al., 2021) and UniHDA (Ours) with the same noise. FHDA and NADA generate images with poor cross-domain consistency, leading to a limited diversity. In contrast, UniHDA alleviates overfitting and maintains strong cross-domain consistency.

| Method | Taylor–Elena | | Hulk–Wooden | | Johnson–Comic | | Average | |
|---|---|---|---|---|---|---|---|---|
| | CS-I (↑) | SCS (↑) | CS-I (↑) | SCS (↑) | CS-I (↑) | SCS (↑) | CS-I (↑) | SCS (↑) |
| FHDA | 0.685 | 0.576 | 0.635 | 0.659 | 0.640 | 0.679 | 0.630 | 0.661 |
| NADA | 0.684 | 0.579 | 0.624 | 0.575 | 0.647 | 0.642 | 0.628 | 0.639 |
| Ours | **0.699** | **0.738** | **0.649** | **0.707** | **0.656** | **0.764** | **0.642** | **0.769** |

Table 1: Quantitative results for **image-image** domain adaptation. We present the results for cases in Fig. 5. To further demonstrate the robustness, we average the results for more cases in Appendix.

## 4 EXPERIMENTS

### 4.1 EXPERIMENTAL SETTINGS

**Methodology.** We demonstrate the versatility of UniHDA on generalized hybrid domain adaptation, *i.e.*, image-image, image-text, and text-text (in Appendix). To show the generator-agnostic nature of UniHDA, we apply it to three well-known generators, *i.e.*, StyleGAN2 (Karras et al., 2020), Diffusion model (Kim et al., 2022a), and EG3D (Chan et al., 2022). Following previous generative domain adaptation literatures (Gal et al., 2021; Zhang et al., 2022; Mo et al., 2020; Li et al., 2020; Nitzan et al., 2023b; Ojha et al., 2021; Zhao et al., 2022b; Xiao et al., 2022; Mondal et al.), we use StyleGAN2 for comparisons in most experiments.

**Datasets.** We conduct experiments for a wide range of source and target domains to validate the effectiveness of UniHDA. Following previous work, we consider FFHQ (Karras et al., 2019), AFHQ-Dog (Choi et al., 2020), and LSUN-Church (Yu et al., 2015) as the source domains. The resolutions of images in these datasets are respectively 1024, 512, and 256. We adapt the generator to diverse hybrid domains driven by the text prompt and one-shot image. To demonstrate the effect of the hybrid domain, *we set the domain coefficients in Eq. (4) as 0.5*. Unless stated otherwise, we use $\lambda = 5$ in Eq. (6) for all experiments.

**Evaluation Metrics.** One important aspect of evaluating generative domain adaptation is the preservation of domain-specific characteristics. Following (Ruiz et al., 2023), we use CLIP Score (CS-T and CS-I) for text-text and image-image adaptation respectively. Concretely, CS-T is measured by the average cosine similarity between the target prompt and generated images' embedding. CS-I is the average pairwise cosine similarity between CLIP embeddings of real and generated images. Here we use the average CS-T or CS-I of multiple domains. For image-text adaptation, we use the average of CS-T and CS-I as the metric (CS). Another important evaluation is the cross-domain consistency of the source domain. To measure it, we adopt the Structural Consistency Score (SCS) (Xiao et al., 2022) to evaluate the spatial structural consistency between the source and target generator.

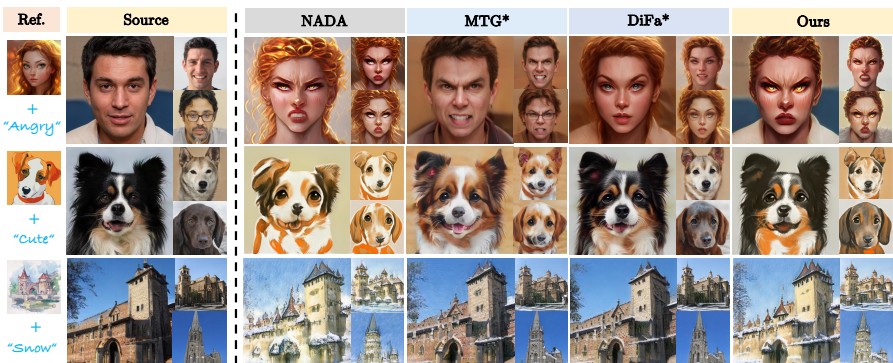

Figure 6: **Image-text** hybrid domain adaptation. We compared our method with previous methods (Gal et al., 2021; Zhu et al., 2021; Zhang et al., 2022), UniHDA well captures the attributes of hybrid target domain and maintains strong cross-domain consistency with source domain. ∗ indicates that MTG and DiFa support multi-modalities by interpolating model parameters with NADA.

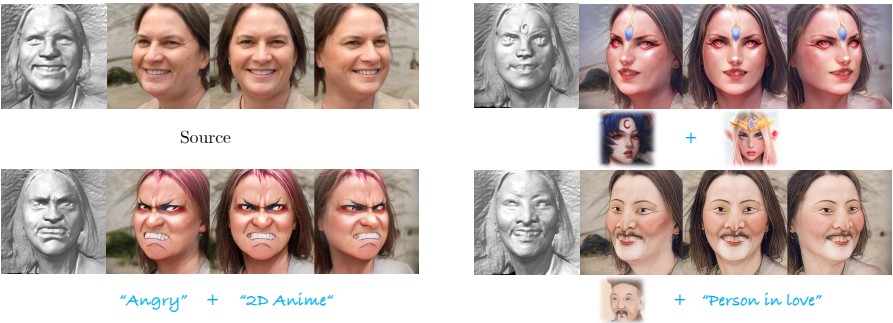

Figure 7: Hybrid domain adaptation in 3D generator. To show the versatility of UniHDA, we apply it on the popular 3D-aware generator, EG3D (Chan et al., 2022).

## 4.2 IMAGE-IMAGE HYBRID DOMAIN ADAPTATION

Fig. 5 shows the qualitative results for image-image adaptation, starting from the same source domain FFHQ (Karras et al., 2019) to the combinations of individual domains. As shown in the figure, FHDA (Li et al., 2023) suffers from severe model collapse and generates images with limited diversity due to the scarcity of image references. While NADA (Gal et al., 2021) mitigates overfitting to a certain extent, its cross-domain consistency remains poor, resulting in the generation of similar images. In contrast, UniHDA maintains strong consistency and effectively generates images with integrated characteristics.

| Method | FFHQ | | Dog | | Church | |
|---|---|---|---|---|---|---|
| | CS (↑) | SCS (↑) | CS (↑) | SCS (↑) | CS (↑) | SCS (↑) |
| NADA | 0.563 | 0.586 | 0.424 | 0.533 | **0.414** | 0.629 |
| MTG | 0.536 | 0.529 | 0.403 | 0.526 | 0.403 | 0.684 |
| DiFa | 0.548 | 0.681 | 0.413 | 0.683 | 0.407 | 0.711 |
| Ours | **0.565** | **0.742** | **0.430** | **0.796** | **0.414** | **0.781** |

Table 2: Quantitative results for **image-text** domain adaptation. We average the results for cases in Appendix.

We also quantitatively compare UniHDA with baselines. As shown in Tab. 1, ours clearly outperforms them. For CS-I, UniHDA significantly outperforms other methods, indicating that generated images effectively integrate multiple characteristics from distinct domains. Furthermore, UniHDA achieves better SCS, which effectively maintains cross-domain consistency compared with baselines.

## 4.3 IMAGE-TEXT HYBRID DOMAIN ADAPTATION

Fig. 6 shows the results of image-text adaptation, including FFHQ (Karras et al., 2019), AFHQ-Dog (Choi et al., 2020), and LSUN -Church (Yu et al., 2015). As depicted in Sec. 4.2, NADA is susceptible to overfitting, which retains poor cross-domain consistency. Besides, we interpolate NADA's parameters with MTG(Zhu et al., 2021) and DiFa (Zhang et al., 2022), which alleviates overfitting to some extent. However, they can't accurately capture the attributes of the hybrid target domain and still fail to maintain good consistency. In contrast, UniHDA well captures the attributes

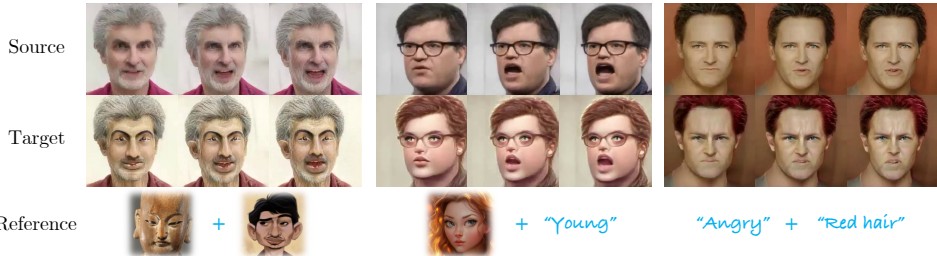

Figure 8: Hybrid domain adaptation in the video generator. Specifically, we apply it to StyleInv (Wang et al., 2023), an unconditional video generator to synthesize high-quality videos.

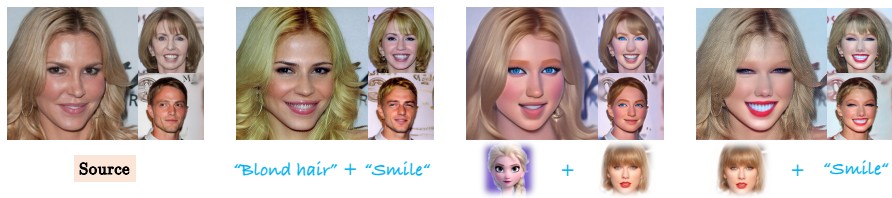

Figure 9: Results of UniHDA with DiffusionCLIP (Kim et al., 2022a), which demonstrate UniHDA is agnostic to the type of generator, allowing for broader application on diffusion models.

and achieves robust consistency in all scenarios. As shown in Tab. 2, we also compare UniHDA with the baselines quantitatively. Consistent with qualitative results in Fig. 6, ours clearly outperforms the baselines. Additionally, we also conduct the user study in the Appendix.

## 4.4 GENERALIZATION ON OTHER GENERATIVE MODELS

In this section, we demonstrate UniHDA is agnostic to the type of generative models and can easily generalize to other generators, *e.g.*, EG3D (Chan et al., 2022), StyleInv (Wang et al., 2023), and DiffusionCLIP (Kim et al., 2022a). For EG3D, a popular 3D-aware image generation method, we replace the discrimination loss with our framework for hybrid domain adaptation. As shown in Fig. 7, the results effectively integrate the attributes and preserve the characters and poses of the source domain. For video adaptation, we conduct experiments on StyleInv (Wang et al., 2023), an unconditional video generator. Fig. 8 verifies the generalization to synthesize high-quality videos. For DiffusionCLIP, we replace the training objective of DiffusionCLIP with our proposed $\mathcal{L}_{direct}$ and $\mathcal{L}_{CSS}$. As shown in Fig. 9, the results integrate the characteristics from multiple target domains and maintain robust consistency with the source domain. More results are included in the Appendix.

## 4.5 COMPARISON WITH TEXT-TO-IMAGE GENERATORS

Recent text-to-image generators like IP-Adapter (Ye et al., 2023) could synthesize promising results with provided image and text prompt. However, the objective of UniHDA is to generate images with attributes of the generalized hybrid target domains while maintaining considerable diversity from the source domain, which can be applicable in scenarios like *data collection*. As shown in Fig. 10, while IP-Adapter can produce images with multiple attributes, their diversity often diminishes when generating large quantities of images. This is due to their inability to retain the distribution of the original domain, which makes it impractical for *data collection*.

## 4.6 COMPARISON WITH IMAGE EDITING

Image editing could somehow yield alike images with the attribute of the target domain. However, UniHDA holds several key differences: (1) Existing editing methods (Patashnik et al., 2021; Duan et al., 2023; Lyu et al., 2023) are based on inversion techniques, which inherently involve the loss of information. Multi-attribute editing exacerbates this problem. As shown in Fig. 11, after multiple edits, the resulting images show poor consistency with the source domain. However, our results maintain strong consistency. (2) UniHDA can adapt generators to more composite and expansive domains, which offers greater flexibility. However, during the editing, subsequent edits often overwrite the previously modified attributes, making it difficult to generate images with a

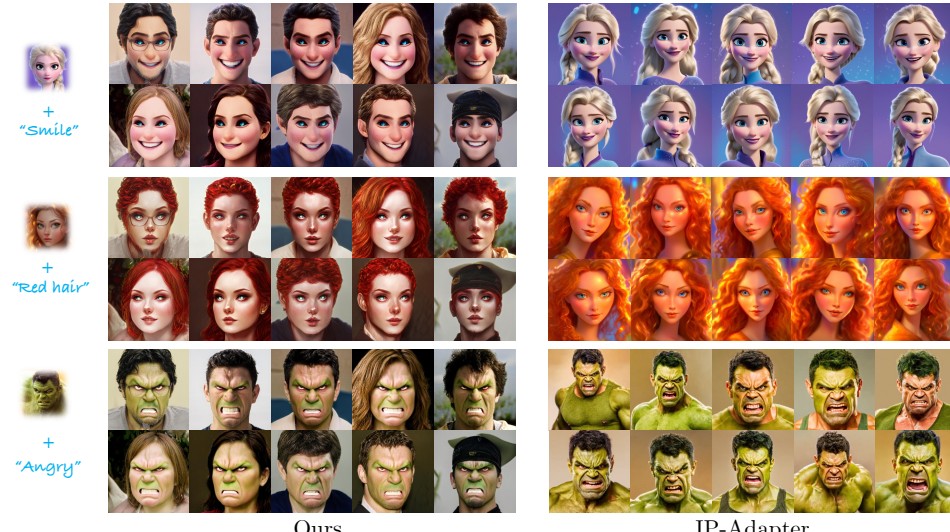

Figure 10: Comparison between UniHDA and IP-Adapter (Ye et al., 2023). Given a single image reference, We adopt the IP-Adapter to generate the results with a fixed text prompt and different random seeds. IP-Adapter tends to overfit the single reference and lose the diversity. Differently, UniHDA maintains robust consistency with the source domain and presents compelling diversity.

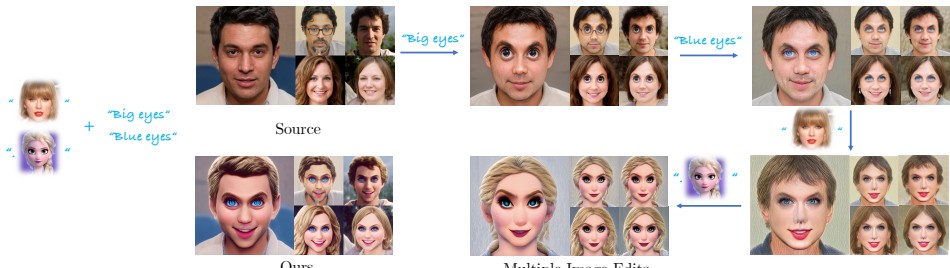

Figure 11: Comparison between UniHDA and image editing. We use StyleCLIP (Patashnik et al., 2021) to perform multiple edits on images for hybrid attributes, which presents two issues: (1) Multiple inversions exacerbate information loss, leading to decreased consistency. (2) Subsequent edits may overwrite the effects of previous edits, as observed that the big eyes attribute disappears after the blue eyes edit.

combination of hybrid attributes as shown in Fig. 11. (3) UniHDA is generator-agnostic and versatile to multiple generators, including 3D generators and video generators. However, 2D image editing is challenging to apply to 3D images or videos because it is difficult to maintain robust multi-view or temporal consistency. (4) The objective of generative domain adaptation is to generate images with attributes of the target domain while maintaining considerable diversity, which can be applicable in scenarios like *data collection*. Image editing, on the other hand, requires the original image as input, rendering it impractical for such applications.

## 4.7 RESULTS OF INCOMPATIBLE DOMAIN ADAPTATION

Typically, the attributes of the target domain and the source domain are complementary. Even in cases where conflicts arise, our UniHDA can maintain robust consistency with the source domain while acquiring the attributes of the target domain. To verify this, we additionally conduct experiments for hybrid domain adaptation on incompatible domains, *i.e.*, from cat to rabbit. As shown in Fig. 12, we start from AFHQ (Choi et al., 2020) cat to incompatible domains. Although there is a conflict between the reference image and the source domain, we can observe that the results still effectively integrate the attributes of the corresponding domain and maintain robust consistency with the source domain.

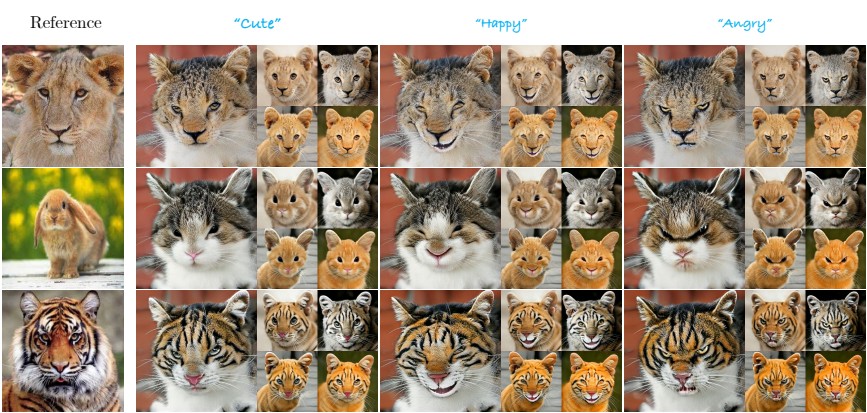

Figure 12: Hybrid domain adaptation from AFHQ-Cat to incompatible domains.

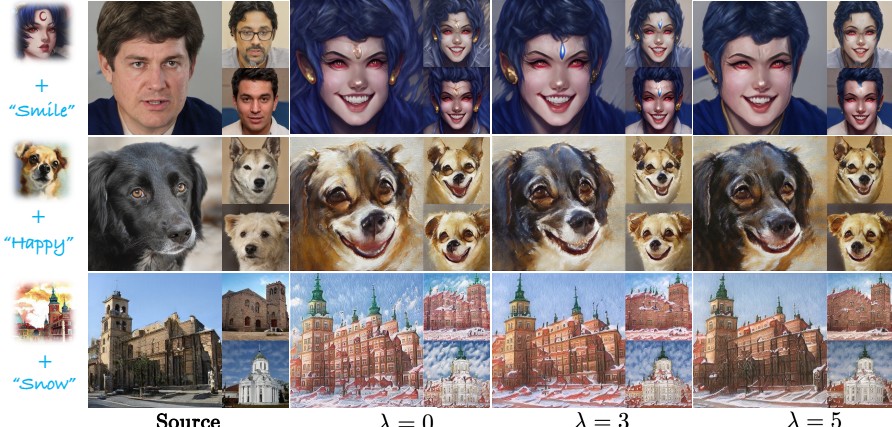

Figure 13: Ablation of our proposed $\mathcal{L}_{\text{CSS}}$ on hybrid domain adaptation, which significantly alleviates overfitting and improves cross-domain consistency. $\lambda$ is the coefficient of $\mathcal{L}_{\text{CSS}}$.

### 4.8 ABLATION OF CSS LOSS

We conduct the ablation study to evaluate the effects of our proposed CSS loss. As shown in Fig. 13, the results without $\mathcal{L}_{\text{CSS}}$ suffer from overfitting and have very limited cross-domain consistency, *e.g.*, distorted backgrounds in Row 1 and 3. Benefiting from $\mathcal{L}_{\text{CSS}}$, the generated images maintain consistency with the source images in terms of spatial structure, thereby inheriting the diversity from the source domain. Besides, we conduct the quantitative ablation in the Appendix. There exists a trade-off between adaptation to the target domain and preserving the characteristics from the source domain. We can adjust the coefficient $\lambda$ based on the desired effect.

## 5 CONCLUSION & LIMITATION

In this paper, we explore a new task, generalized hybrid domain adaptation, and propose UniHDA, a unified and versatile framework to enable it. For the hybrid domain, we demonstrate the compositional capabilities of direction vectors in CLIP's embedding space and linearly interpolate direction vectors of multiple target domains. We also propose a new cross-domain spatial structure loss to improve consistency, which is conducted in generator-agnostic space and versatile for various generators. We believe our work is an important step towards generative domain adaptation, since we have demonstrated the source generator can be effectively adapted to a hybrid domain with multi-modal references and maintain robust cross-domain consistency. Our code will be made public.

While UniHDA effectively realizes generalized hybrid domain adaptation, it also has limitations. To encode both image and text into a shared embedding space, we utilize pre-trained CLIP during training time, which might bring potential bias for some domains. Nevertheless, we believe that the exploration of the novel task is significant for future work and solutions could be integrated into UniHDA to eliminate the bias.

## 6 ETHICS STATEMENT

Our main objective in this work is to empower novice users to generate visual content creatively and flexibly. However, the broad adoption of such technology brings up ethical issues related to privacy, misinformation, and potential misuse. We strongly support the responsible development and deployment of tools to detect biases and malicious use cases, highlighting the need for ethical standards to guarantee their safe and ethical use in the field of computer vision.

## 7 REPRODUCIBILITY STATEMENT

We make the following efforts to ensure the reproducibility of UniHDA: (1) Our training and inference codes together with the trained model weights will be publicly available. (2) We provide the details of the human evaluation setups in the appendix (Appendix A.5). (3) We provide training details in the appendix (Appendix A.8), which is easy to follow.

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

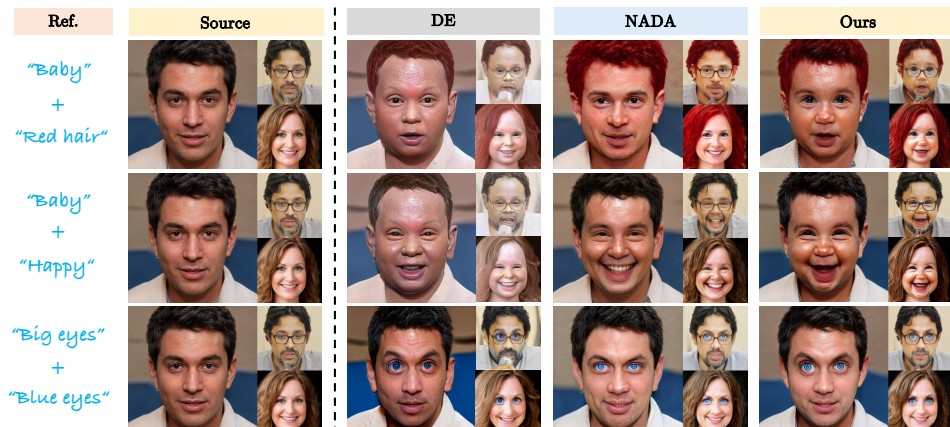

Figure 14: **Text-text** hybrid domain adaptation. We compare the results of DE (Nitzan et al., 2023b), NADA (Gal et al., 2021) and UniHDA (Ours) with the same noise. UniHDA exhibits desirable performance to acquire characteristics from hybrid target domain and maintain robust cross-domain consistency.

| Method | Baby-Red hair | | Baby-Happy | | Big-Blue eyes | | Average | |
|---|---|---|---|---|---|---|---|---|
| | CS-T (↑) | SCS (↑) | CS-T (↑) | SCS (↑) | CS-T (↑) | SCS (↑) | CS-T (↑) | SCS (↑) |
| DE | 0.163 | 0.638 | 0.160 | 0.580 | 0.195 | 0.662 | 0.167 | 0.634 |
| NADA | 0.179 | 0.661 | 0.170 | 0.642 | 0.186 | 0.731 | 0.159 | 0.552 |
| Ours | **0.186** | **0.744** | **0.175** | **0.757** | **0.197** | **0.765** | **0.176** | **0.707** |

Table 3: Quantitative results for **text-text** domain adaptation. We present the quantitative results corresponding to each case in Fig. 14. Similar to Tab. 1, we average the results for cases in Appendix.

# A APPENDIX

In this appendix, we begin to conduct the experiments on text-text hybrid domain adaptation in Appendix A.1. And we compare with existing methods in terms of efficiency in Appendix A.2. Then we show more ablation of CSS loss in Appendix A.3 and Appendix A.4, including quantitative results and the effect of the encoder in CSS loss. Additionally, we report the user study in Appendix A.5. Then we show more qualitative results *e.g.*, more domains in Appendix A.6, and additional results in Appendix A.7. Finally, we provide more implement details in Appendix A.8 and the potential bias in Appendix A.9.

## A.1 TEXT-TEXT HYBRID DOMAIN ADAPTATION

Fig. 14 shows the qualitative results for text-text adaptation. Since the adaptation is conducted solely along one projection direction of the latent code, Domain Expansion (DE) (Nitzan et al., 2023b), does not fully capture the characteristics of the target domain, *e.g.*, baby (Row 1 and Row 2). Furthermore, DE does not maintain robust consistency, *e.g.*, the chin of the person in the upper-right corner and background artifacts in Row 3. The problem of NADA (Gal et al., 2021) is overfitting. Hard-to-learn characteristics, *e.g.*, baby (Row 1 and Row 2) and big eyes (Row 3) may be overshadowed by other overfitted ones. In contrast, UniHDA (Ours) exhibits desirable performance to generate images with integrated characteristics while maintaining robust consistency with the source domain.

Similar to Sec. 4.2, we also compare UniHDA with the baselines quantitatively. As shown in Tab. 3, ours clearly outperforms the baselines, which are consistent with qualitative results in Fig. 14. We achieve better CS-I and SCS, indicating that generated images effectively integrate domain-specific attributes and preserve primary characteristics of the source domain.

| Method | Modality | Model Amount | 2-domain size (↓) | 2-domain time (↓) | 10-domain size (↓) | 10-domain time (↓) |
|---|---|---|---|---|---|---|
| NADA | Multi | $N$ | 48M | 4min | 240M | 20min |
| MTG* | Multi | $N$ | 48M | 4min | 240M | 20min |
| DiFa* | Multi | $N$ | 48M | 4min | 240M | 20min |
| DE† | Text | 1 | 24M | 20h | 24M | 20h |
| FHDA | Image | 1 | 24M | 3min | 24M | 3min |
| Ours | Multi | 1 | **24M** | **2min** | **24M** | **2min** |

Table 4: Comparison with previous methods. ∗ indicates MTG and DiFa support multi-modalities by interpolating model parameters with NADA. † means DE needs source dataset (e.g., FFHQ) that significantly increases training time.

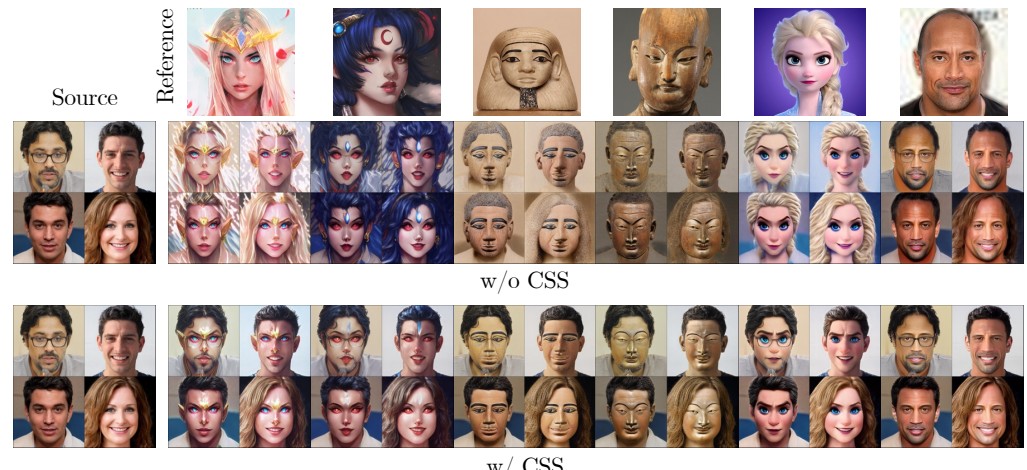

Figure 15: More qualitative results to verify the effectiveness of our proposed $\mathcal{L}_{\text{CSS}}$.

## A.2 COMPARISON WITH EXISTING METHODS

In addition to generation quality, UniHDA also surpasses existing methods in terms of efficiency, *e.g.*, model size and training time as shown in Tab. 4. NADA, MTG, and DiFa trains a separate generative model per domain and interpolates their parameters in test-time, which necessitates multiple times the model size and training time. Although DE avoids cross-model interpolation, it heavily relies on the large source dataset for regularization during training process, resulting in a significant increase in training time. In contrast, UniHDA circumvents these issues, which enables the adaptation within single generator in only two minutes.

Furthermore, DE relies on the semantic latent space of the generator (*e.g.*, StyleGAN (Karras et al., 2019) and DiffAE (Preechakul et al., 2022)) for hybrid domain adaptation, limiting its applicability to a broader range of generators. MTG and DiFa utilize GAN inversion, which restricts the applicability to generators similar to StyleGAN. Conversely, UniHDA is not constrained by the type of generators, allowing for its broader application across various generators.

## A.3 MORE ABLATION OF CSS LOSS

As depicted in Sec. 4.8 of the main paper, our proposed $\mathcal{L}_{\text{CSS}}$ significantly alleviates overfitting and improves cross-domain consistency. Results with $\mathcal{L}_{\text{CSS}}$ achieve better SCS score, indicating that they maintain stronger consistency with the source domain. Additionally, we show more qualitative results in Fig. 15 to verify the effectiveness of UniHDA. We also conduct the quantitative ablation in Tab. 5, which is consistent with the qualitative results.

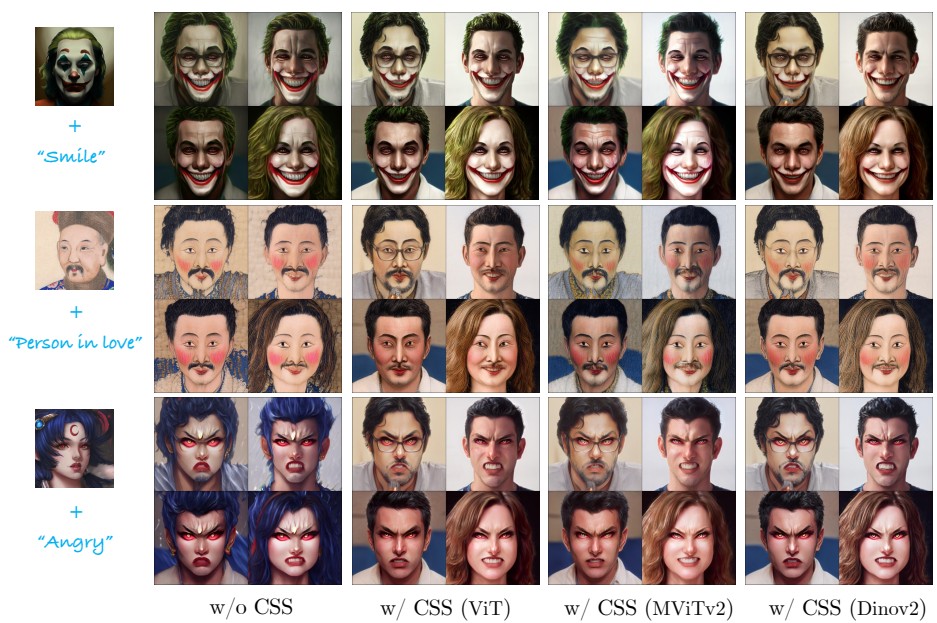

Figure 16: Ablation of different pre-trained encoders for CSS on hybrid domain adaptation.

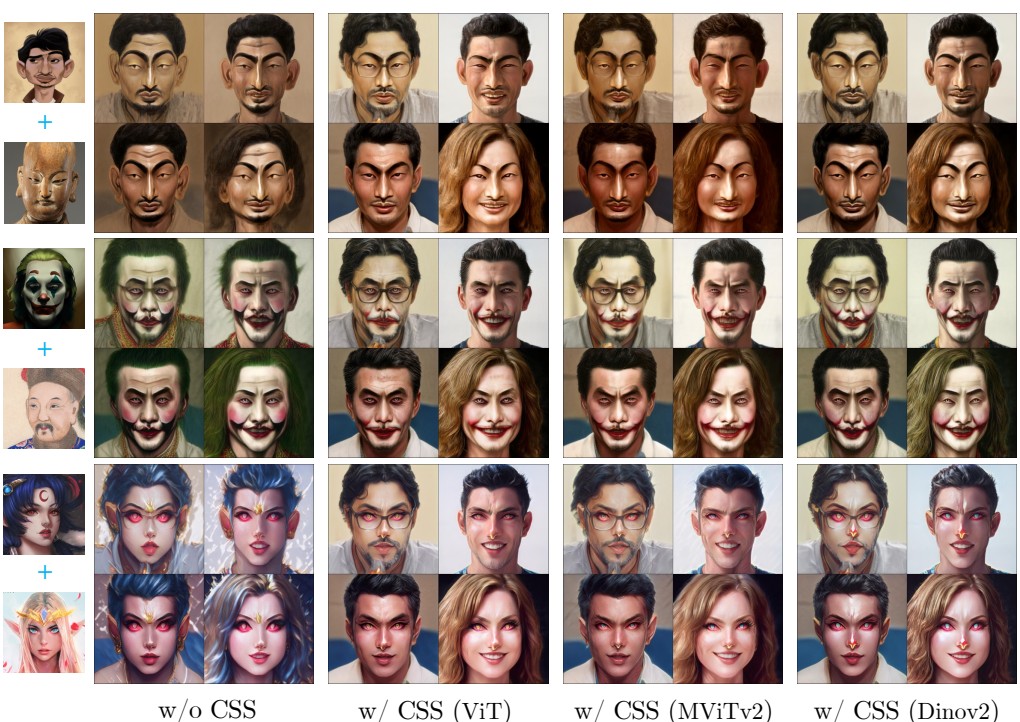

Figure 17: Effect of different pre-trained image encoders for CSS on image-image hybrid domain adaptation.

### A.4 ABLATION OF ENCODER FOR CSS

We conduct experiments on pre-trained ViT (Dosovitskiy et al., 2020), MViTv2 (Li et al., 2022), and Dinov2 to explore the impact of different image encoders for CSS. As shown in Fig. 16, Fig. 17 and

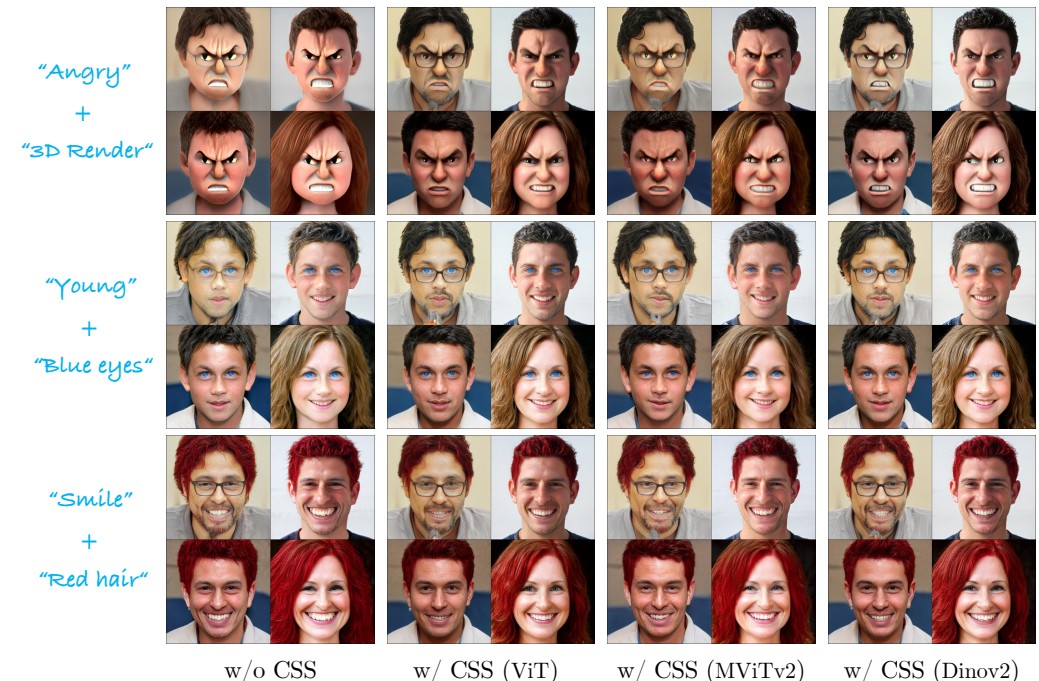

"Angry"
+
"3D Render"

"Young"
+
"Blue eyes"

"Smile"
+
"Red hair"

| w/o CSS | w/ CSS (ViT) | w/ CSS (MViTv2) | w/ CSS (Dinov2) |

Figure 18: Effect of different pre-trained image encoders for CSS on text-text hybrid domain adaptation.

| $\lambda$ | FFHQ (I-I) | | FFHQ (T-T) | | FFHQ (T-I) | | Dog (T-I) | | Church (T-I) | |
|---|---|---|---|---|---|---|---|---|---|---|
| | SCS (↑) | CS-I (↑) | SCS (↑) | CS-T (↑) | SCS (↑) | CS (↑) | SCS (↑) | CS (↑) | SCS (↑) | CS (↑) |
| 0 | 0.502 | 0.639 | 0.520 | 0.170 | 0.562 | 0.557 | 0.491 | **0.430** | 0.604 | 0.411 |
| 3 | 0.681 | 0.638 | 0.683 | 0.171 | 0.694 | 0.556 | 0.787 | 0.428 | 0.706 | 0.413 |
| 5 | **0.769** | 0.642 | 0.707 | **0.176** | 0.742 | **0.565** | 0.796 | **0.430** | **0.781** | **0.414** |
| 10 | 0.762 | **0.655** | **0.756** | 0.170 | **0.773** | 0.514 | **0.798** | 0.425 | 0.778 | 0.410 |

Table 5: Quantitative ablation for our proposed $\mathcal{L}_{\text{CSS}}$. $\lambda$ is the coefficient of $\mathcal{L}_{\text{CSS}}$. There exists a trade-off between adaptation to the target domain and preserving the characteristics from the source domain. We can adjust $\lambda$ based on the desired effect.

Fig. 18, we can observe that all of them improve the consistency with source domain compared with the baseline approach. Furthermore, they exhibit a similar qualitative style, which demonstrates that our CSS is agnostic to different pre-trained image encoders.

| Method | Fidel. | Diver. | Corr. |
|---|---|---|---|
| vs. NADA (I-I) | 85.2 | 90.6 | 76.0 |
| vs. NADA (T-T) | 81.4 | 84.2 | 80.8 |
| vs. NADA (T-I) | 84.6 | 85.8 | 78.6 |

Table 6: User study for fidelity, diversity, and reference correspondence (image or text) in hybrid domain adaptation. The value (%) represents the percentage of users who favor the images generated by our method over NADA.

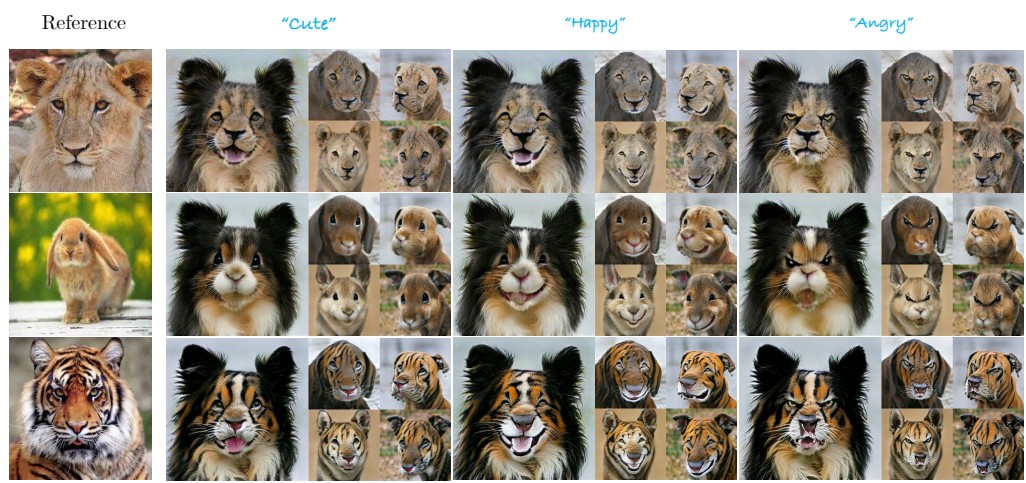

Figure 19: Hybrid domain adaptation from AFHQ-Dog to incompatible domains, *i.e.*, lion, rabbit, and tiger.

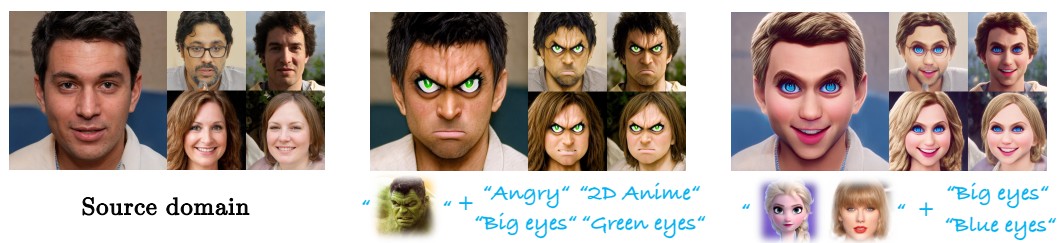

Figure 20: The results of hybrid domain adaptation from FFHQ to the hybrid of more domains.

### A.5 USER STUDY

We conduct user study in Tab. 6 to compare UniHDA with NADA. Specifically, we provide users with the target text or image, the source image, and adapted images. Then we ask them to choose the better image for fidelity, diversity and correspondence. For each case, we generate 1000 samples and randomly assign 200 samples to 30 users. The results indicate that UniHDA surpasses NADA in terms of fidelity, diversity and reference correspondence.

### A.6 MORE QUALITATIVE RESULTS

We apply UniHDA to adapt the generator on FFHQ (Karras et al., 2019) to more hybrid domains, *i.e.*, text-text, image-image, and image-text, as well as AFHQ (Choi et al., 2020) dog to incompatible domains. As shown in Fig. 20, Fig. 21, Fig. 22, Fig. 23, and Fig. 19. UniHDA successfully generates images with integrated characteristics from multiple target domains and maintains robust consistency with the source domain. Besides, we showcase more results of hybrid domain adaptation from AFHQ-Dog and LSUN-Church (Yu et al., 2015) in Fig. 24 and Fig. 25.

### A.7 MORE RESULTS FOR DIFFUSIONCLIP AND EG3D

To demonstrate the versatility of UniHDA, we apply it on DiffusionCLIP and EG3D in Sec. 4.4 in the main paper. As shown in Fig. 26 and Fig. 27, we showcase more results including image-image, text-text, and image-text. All results achieve hybrid domain adaptation and preserve strong cross-domain consistency.

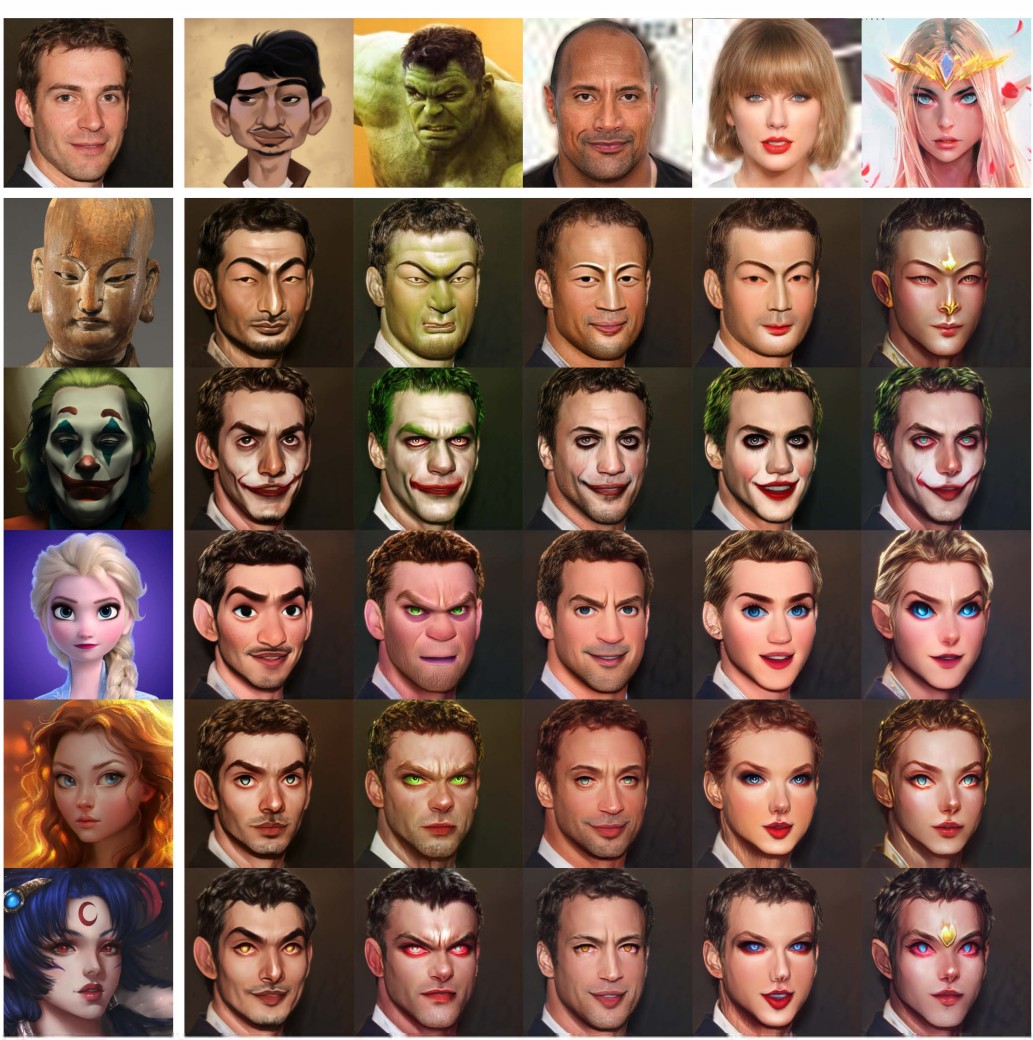

Figure 21: More results of image-image hybrid domain adaptation. The source image is in the top-left corner, and the first row and column consist of training images.

### A.8    IMPLEMENT DETAILS

Following the setting of previous generative domain adaptation methods (Gal et al., 2021; Nitzan et al., 2023b), we utilize the batch size of 4 and ADAM Optimizer with a learning rate of 0.002 for all experiments during training. A training session typically requires 300 iterations in 2 minutes, which significantly reduces training time compared with adversarial methods for generative domain adaptation. Note that we conduct all experiments on a single NVIDIA RTX 4090 GPU. The code will be open source.

For experiments on FFHQ, we generate images with 1024 × 1024 resolution. As for AFHQ-Dog and LSUN-Church, we operate on 512 × 512 and 256 × 256 resolution images respectively.

### A.9    POTENTIAL BIASES OF CLIP

As depicted in NADA, CLIP may introduce textual bias and ambiguity in some specific domains. For example, the text 'Nurse' tends to convert the individuals to females, as shown in Fig. 28.

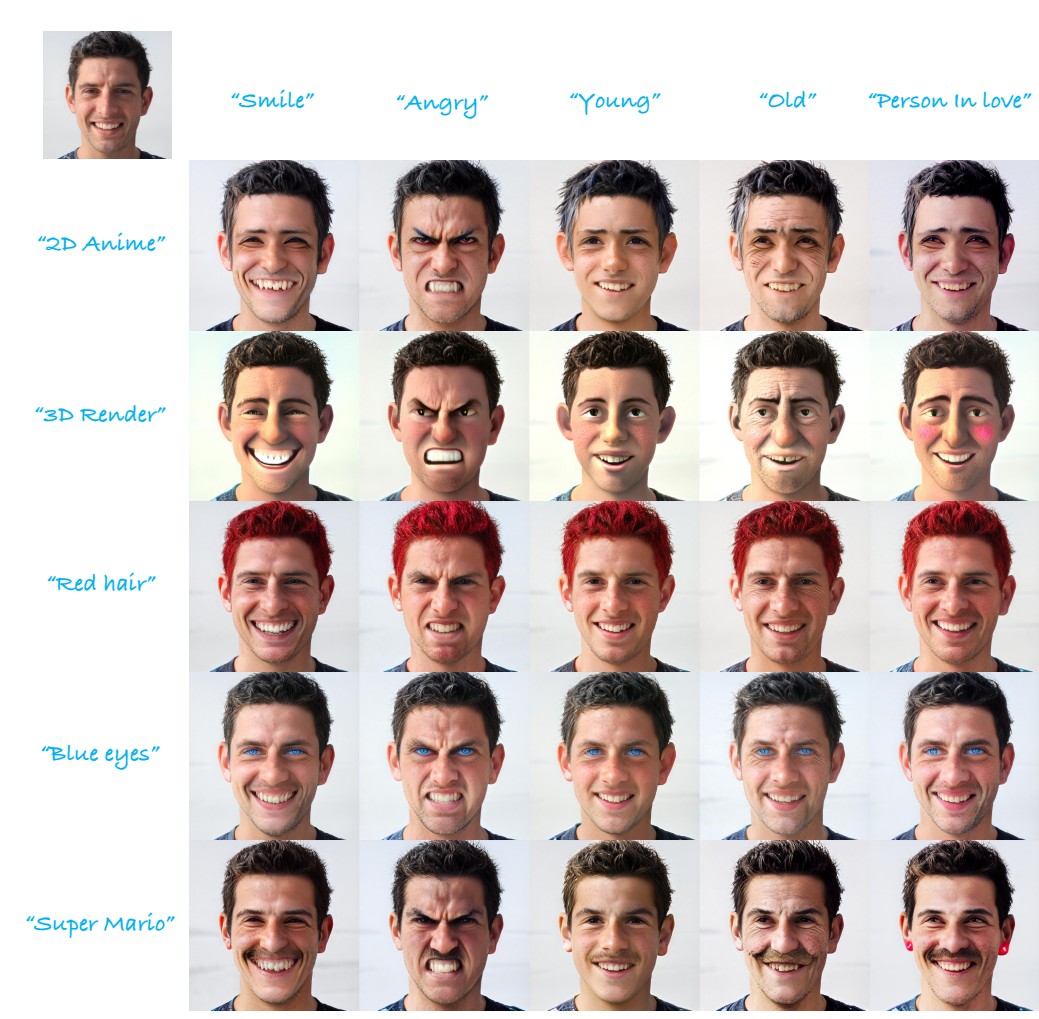

Figure 22: More results of text-text hybrid domain adaptation. The source image is in the top-left corner, and the first row and column consist of text prompts.

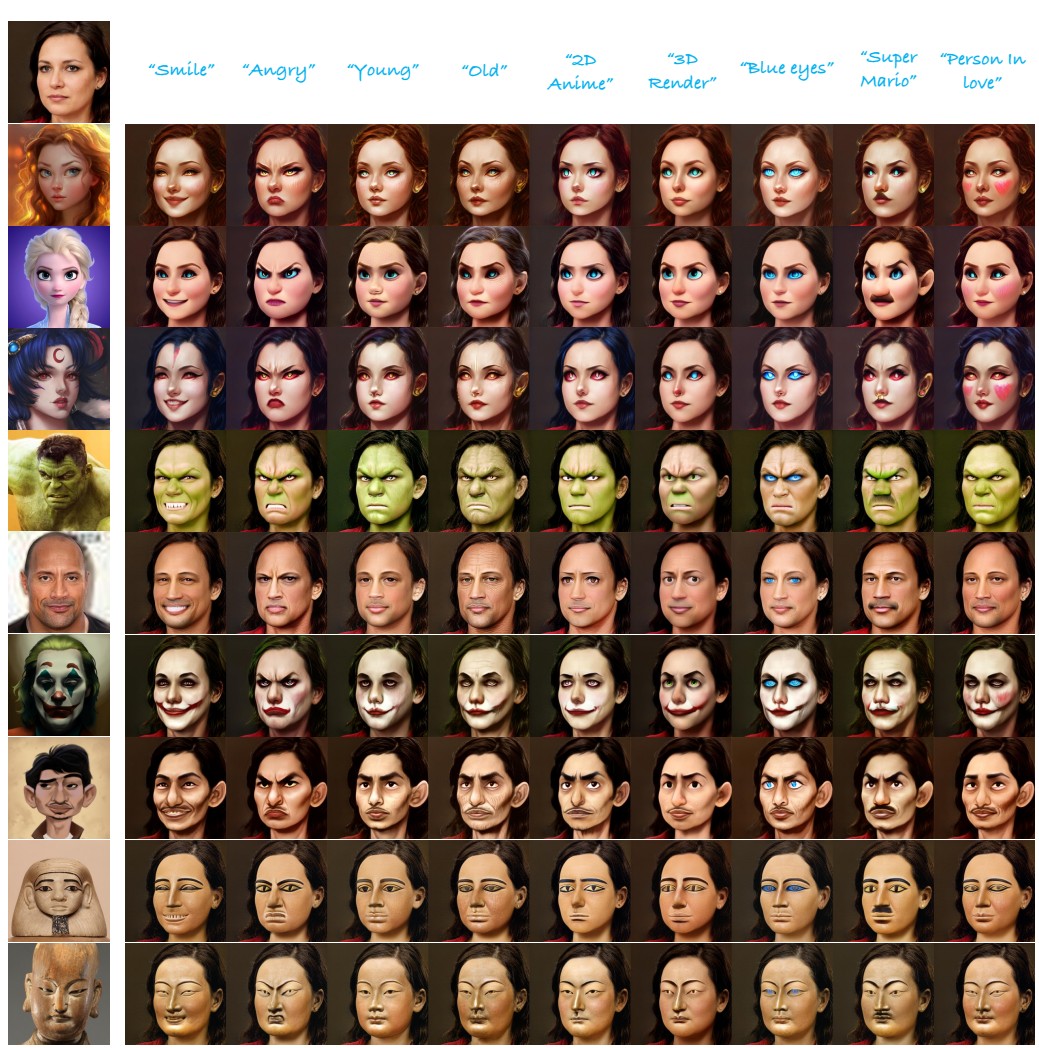

Figure 23: More results of image-text hybrid domain adaptation. The source image is in the top-left corner. The first row and column consist of training images and text prompts respectively.

Figure 24: More results of image-text hybrid domain adaptation on AFHQ-Dog.

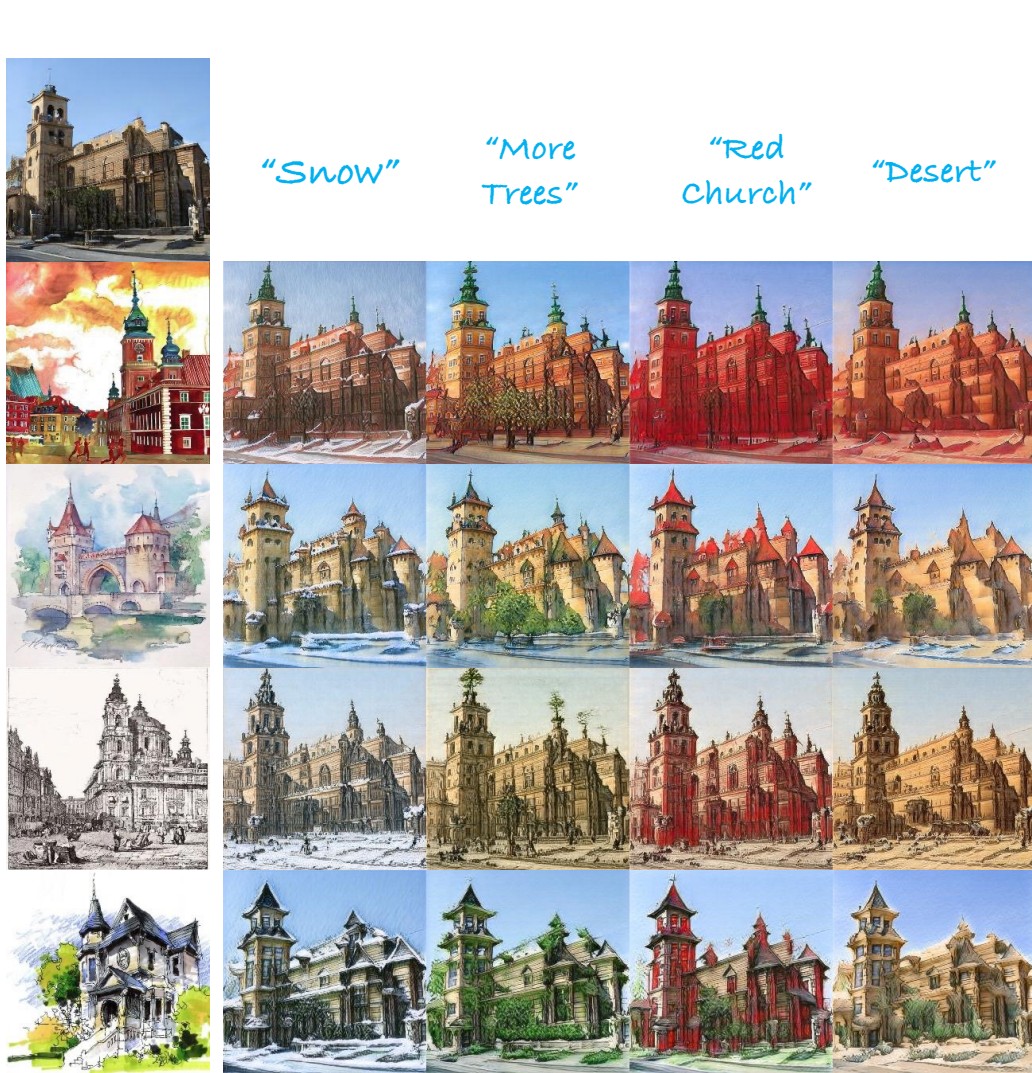

Figure 25: More results of image-text hybrid domain adaptation on LSUN-Church.

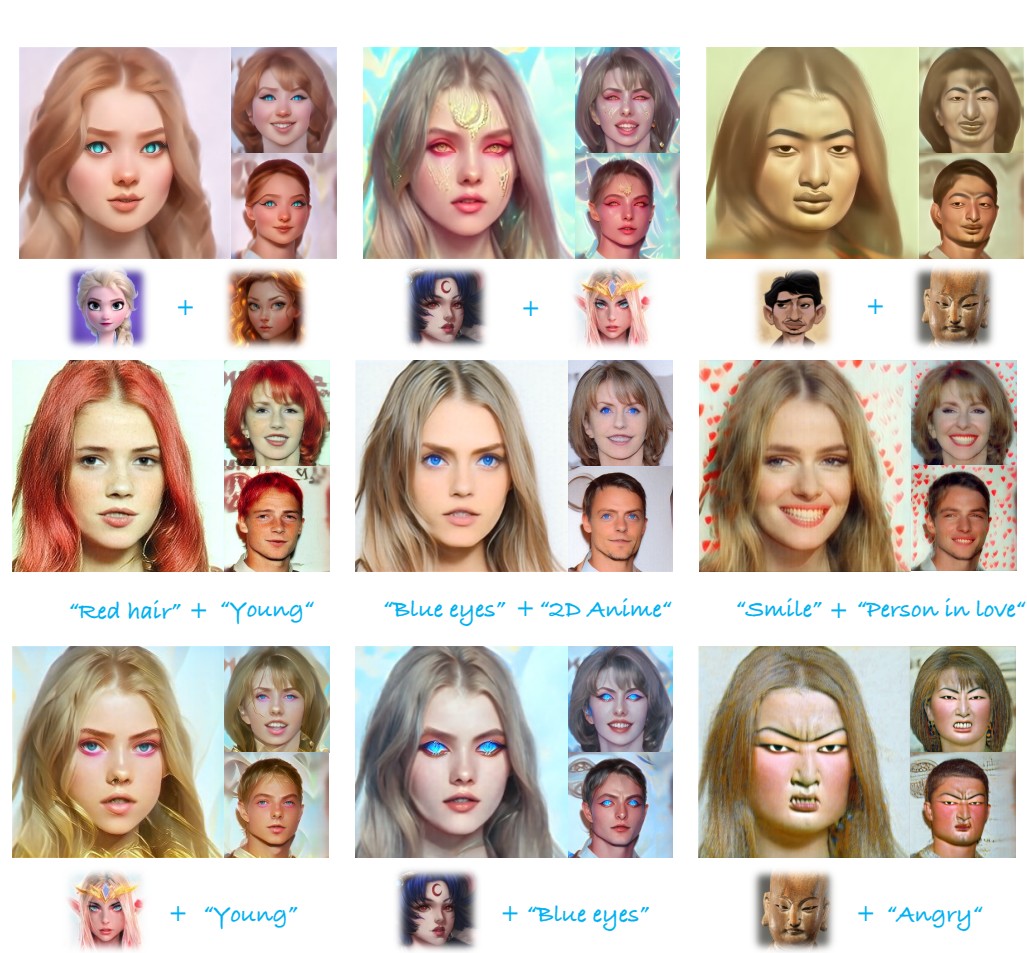

Figure 26: More results of UniHDA with DiffusionCLIP.

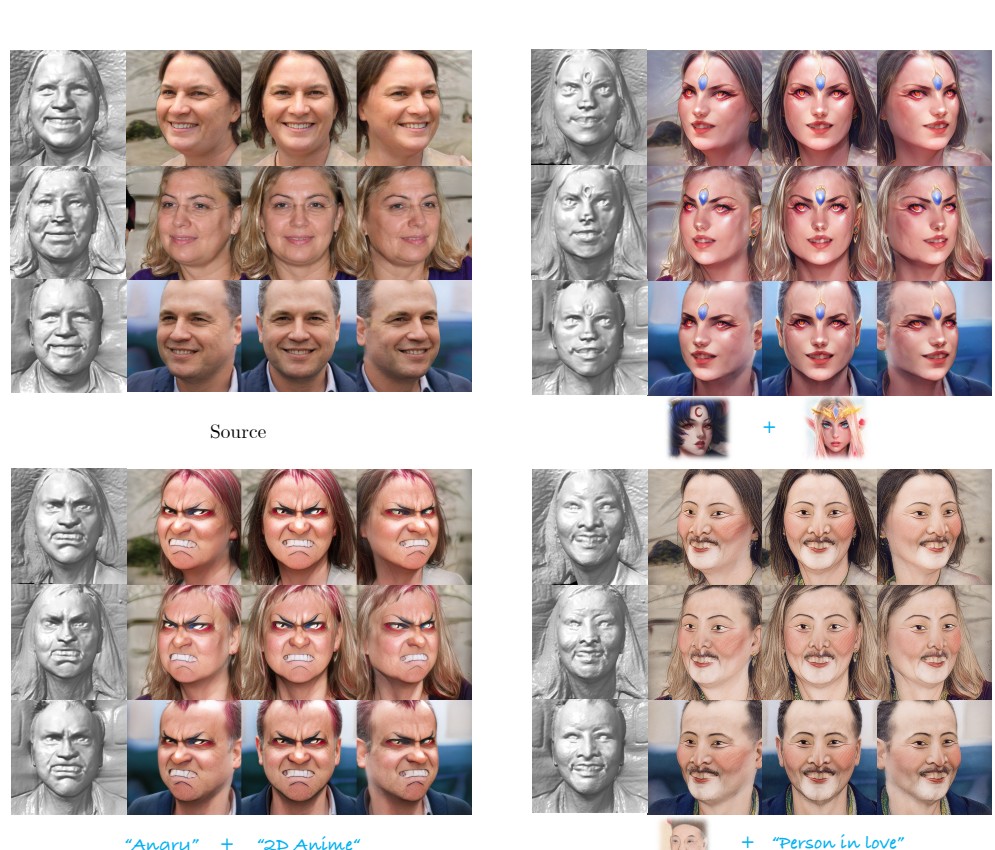

Figure 27: More results of UniHDA with EG3D.

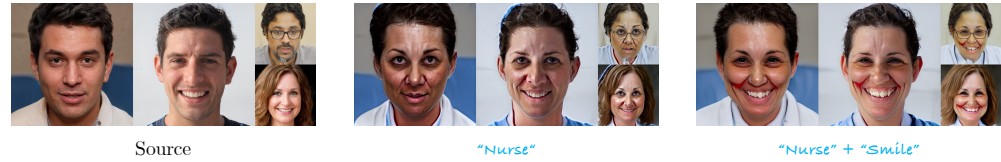

Figure 28: Textual bias and ambiguity introduced by CLIP. We use 'Nurse' as the target domain and CLIP's learned biases manifest in the new domain, which converts the individuals to females.

