# OpenReview forum: "UniHDA: A Unified and Versatile Framework for Generalized Hybrid Domain Adaptation"
_ICLR.cc/2025/Conference — ICLR 2025 Conference Withdrawn Submission_

### Official Review · Reviewer_XMAr · 2024-11-01

**Soundness:** 2
**Presentation:** 3
**Contribution:** 2
**Rating:** 5
**Confidence:** 4

**Summary:**

This paper introduces UniHDA for generalized hybrid domain adaptation, addressing the challenge of adapting pre-trained generators to multiple target domains using multi-modal inputs. UniHDA leverages linear interpolation in CLIP’s embedding space to achieve adaptation across multiple domains. The authors also propose a Cross-Domain Spatial Structure (CSS) loss to preserve fine spatial details of source images. The framework is generator-agnostic and works with various generators like StyleGAN, EG3D, and video generators.

**Strengths:**

1. The paper generally is well-written and easy to follow.
2. Both quantitative and qualitative results demonstrate that UniHDA surpasses prior arts in hybrid domain adaptation.
3. UniHDA validates its effectiveness on 3D and video generator.

**Weaknesses:**

1. The claim that UniHDA is more efficient than NADA’s model interpolation is only partially accurate. As shown in Figure 3, UniHDA requires training six different models to achieve results across various coefficients, whereas NADA only needs two models, relying on weight interpolation between them.
2. The authors’ claim overlooks a more general setting for hybrid domain adaptation. For n target domains, any combination of k domains (where k < n) with arbitrary coefficients could form a potential hybrid. UniHDA, however, demands separate training for each hybrid configuration, whereas NADA only requires training n models, followed by interpolation of their weights based on the desired coefficients.
3. The comparison with IP-adapter seems problematic. IP-adapter functions as an open-set image variation model and lacks the concept of a source domain, making it unsuitable for direct comparison with generative domain adaptation task addressed by UniHDA.

**Questions:**

1. What is the performance of NADA's model interpolation with CSS regularization? This is a crucial ablation study to assess the impact of direction interpolation on the final performance.
2. How are domain coefficients determined for hybrids involving more than two domains?

---

### Official Review · Reviewer_REEr · 2024-11-02

**Soundness:** 3
**Presentation:** 3
**Contribution:** 2
**Rating:** 3
**Confidence:** 4

**Summary:**

The paper proposes Generalized Hybrid Domain Adaptation that claims to adapt existing generators including StyleGAN2 and Eg3D with
hybrid target domain and multi-modal references. The core observation made by the authors is the strong compositional capabilities of direction vectors in CLIP’s embedding space. These directions can be  linearly interpolated for generalized
hybrid domain adaptation. The paper also proposes a cross-domain spatial structure loss to maintain consistency with the source domain. The paper claims that the method is generator agnostic.

**Strengths:**

1) The paper claims that the method is generator agnostic. The analysis of domain adaptation is done in the CLIP embedding space and applied to different generators. There are examples of the same shown in the paper.

2) The results show that the features from a reference image or text are transferred to the target domain. The method is able to mix and match these modalities in the examples shown.

3) The paper compares with the existing StyleGAN based methods and show the shortcomings of some of these methods.

**Weaknesses:**

1) My main concern is regarding the quality of the results. Looking at the videos provided with the suppl. the quality of the videos are not nice. Judging by the quality that eg. Eg3D can provide with the domain adaptation methods e.g (3D avatarGAN https://rameenabdal.github.io/3DAvatarGAN/ ), the geometry of the heads are flattened and the texture quality of the target generator degrades.

2) Applied to the real images (Figure 8), there is obvious blurriness and artifacts on the face.

3) Figure 10 Hulk example seems a bit off. The methods seems to capture the expressions but fails to capture the texture. What if the latents are adapted to match the skin color of hulk. Does it produce visible artifacts?

**Questions:**

1) I would like the authors to comment on the quality of the samples produced by the method related to point 1) in the weakness section.

2) Why are the real face domain adapted results blurry? How can this be mitigated?

3) Some of the domain adapted results do not match the reference image i.e hulk example, why is this the case? Are there visible artifacts when the domain is further shifted.

4) How many consecutive mix of domains can be performed before the model breaks?

---

### Official Review · Reviewer_eH6K · 2024-11-03

**Soundness:** 3
**Presentation:** 3
**Contribution:** 2
**Rating:** 6
**Confidence:** 4

**Summary:**

The paper addresses limitations in existing domain adaptation methods that rely on either image or text conditioning. The authors propose a novel multi-target approach that enables simultaneous adaptation across both image and text modalities. Notably, they identify that using multiple targets can compromise consistency with the source domain, leading them to introduce a spatial structure loss to mitigate this issue.

**Strengths:**

- The paper demonstrates good clarity and readability, with a well-structured presentation of ideas.
- The proposed method is effective and simple, showing versatility to be applied to different models while maintaining performance.
- The approach shows robust performance on incompatible domains, avoiding artifacts that often appear on other methods when objectives are misaligned.
- The authors provide discussion of related approaches, particularly in distinguishing their approach from T2I models (Sec 4.5) and image editing methods (Sec 4.6), which helps readers understand the paper's unique contributions.

**Weaknesses:**

- The experimental validation is limited to two-domain scenarios, leaving the method's scalability to multiple domains unexplored.
- The authors make an unsubstantiated claim (Line 420) regarding their method's superior diversity compared to T2I generators for multi-attribute images. This assertion requires quantitative evidence.
- The discussion of related work overlooks several recent and relevant diffusion-based methods, including attribute manipulation via linear directions [1,2], conditional input guidance [3], target prompt guidance [4,5,6].


[1] Prompt Sliders for Fine-Grained Control, Editing and Erasing of Concepts in Diffusion Models, ECCV 2024.
[2] Concept Sliders: LoRA Adaptors for Precise Control in Diffusion Models, ECCV 2024
[3] Sdedit: Image synthesis and editing with stochastic differential equations, ICLR 2022
[4] Zero-shot Image-to-Image Translation, SIGGRAPH 2023
[5] Prompt-to-Prompt Image Editing with Cross-Attention Control, arXiv 2022
[6] Imagic: Text-Based Real Image Editing with Diffusion Models, CVPR 2023

**Questions:**

Have authors considered enhancing the composition of direction vectors through non-linear interpolation or learnable domain coefficients? What are your thoughts on the potential benefits versus the added complexity of such approaches?

---

### Official Review · Reviewer_qjXb · 2024-11-04

**Soundness:** 2
**Presentation:** 2
**Contribution:** 2
**Rating:** 5
**Confidence:** 4

**Summary:**

The paper tackles the problem of hybrid domain adaptation for image generation. Unlike typical domain adaptation works that adapts source domain to a single target domain, the advantage of the proposed framework lies at an ability to adapt to diverse domains, described either by images or texts. To allow multi-modal (image and text) modality adaptation, authors extended direction loss to both image and text CLIP embedding spaces. Furthermore, authors proposes linear combination of domain-shift direction vectors to compute the direction loss. In addition, authors proposes a cross-domain spatial structure loss to retain structural / spatial consistency. Experiments are done on wide range of generators including GAN, diffusion, 3D, etc.

**Strengths:**

* The proposed method is described well and technically sound. While the direction loss has been already studied for text-guided image editing from StyleGAN-NADA, the multi-modal (combinations of image and text) guidance for image editing seems new.

* Thorough experiments using wide range of image generation models.

**Weaknesses:**

* Visualization of generated images are relatively low quality comparing to state-of-the-art text-to-image generation models of today.

* Improvement over previous work is not very clear from the visual results. For example, in Figure 5 second and third rows, NADA results are better representation of the styles in the reference images than the proposed method (e.g., green color + wooden texture in the second row are better represented by NADA; exaggerated nose and ears are better represented by NADA in the third row).

* In line 319: CS-I is the average pairwise cosine similarity between CLIP embeddings of real and generated images --> is the **real** image referring to the source image? Or the reference image of the target domain? If it is the source image, numbers reported in Table 1, CS-I (cosine similarity between source and generated images) and SCS (structural consistency score between source and target generator), are both maximized when target generator simply outputs the source image. Please clarify.

* Related to the above comments, I wonder the high CS-I and SCS scores of the proposed method is obtained due to the high $\lambda$ value. Could authors provide comprehensive ablation of $\lambda$ (CSS loss) between 0 (no CSS loss) to 5 (value used in the paper) with CS-I and SCS scores as in Table 1?

* Comparison to IP-adapter does not make much sense and could be misleading. Besides the needs for the data collection, IP-adapter is meant to be used to generate variations of the reference image (e.g., variations of the subject in the reference image) with the help of variations of text prompts.

**Questions:**

See also weakness.

* Is the method applicable to state-of-the-art text-to-image generation models of today to achieve better performance? The quality of generated images is low in today's standard and would be beneficial if method can be proven useful for state-of-the-art diffusion models.

* Can method be applied to more than 2 target domains? Most results shown in the paper only consider hybrid domains of two (image-image, text-text or image-text), while the formulation in Eq (4) suggests method works with more than two target domains.

* line 205: it is unclear how $\bar{f}_{s}$ are derived. Are images randomly generated from $G_s$ without any constraint?

* line 318: is CS-T for text-text or image-text?

**Details Of Ethics Concerns:**

* The paper presents several visual results of realistic human faces without clear reference.

---

### Official Review · Reviewer_bNDW · 2024-11-11

**Soundness:** 3
**Presentation:** 3
**Contribution:** 2
**Rating:** 5
**Confidence:** 5

**Summary:**

This paper proposes a generative model adaptation scheme that can adapt to a composition of target domains using either text description or a single image representing each domain. This work has two major contributions: i) enabling adaptation using both text and image data with a simple linear interpolation (more like summation) between the offset of multiple domains, ii) proposing a cross-domain regularizer to prevent overfitting to target domain that suffers from scarcity of the available samples (zero-shot or one-shot scenario).

The idea is simple and interesting, but there are some similarities to prior works and the paper needs to address the prior works more accurately. In addition, experimental results need to be improved to reflect the performance of the proposed method compared to previous approaches.

**Strengths:**

Even though not surprising, the fact that interpolation of directions from multiple domains (in the forms of either text offset or image offset) can lead to the direction of the multiple domains is interesting.

The paper is written well and easy to follow.

**Weaknesses:**

I have some concerns regarding the current version of the paper and I believe addressing these concerns can improve the paper:

$ $

1. **Some claims in the paper are either non-accurate or do not have enough supporting evidence.** For examples:

- (line 86) *'With multiple target domains and very limited references from each domain, the generator is more prone
to overfitting domain-specific attributes'* $\rightarrow$ There is no evidence or empirical study to support this. For example, how adapting to two target domains (one described with image (one-shot) and another described with text (zero-shot)) is more prone than adapting a generator to a single domain that is described only by text (zero-shot).

- (line 188) *'Despite the promising results of existing methods, a major limitation of them is that they only support adaptation from the source domain to individual target domains ...'* $\rightarrow$ Adapting to multiple domains is previously addressed in domain re-modulation [1], FHDA [2] and Domain Expansion [3] papers. However, I believe that conditioned on the input modality, to the best of my knowledge, this work is the first one to address that.

$ $

2. **Writing can be improved to address previous works more accurately and also in some parts prevent cofusion.** More specifically:

- Section 3.2: The details discussed here are basically NADA loss for zero-shot or one-shot adaptation. Similarly, MindTheGap paper [4] extends the NADA idea to the one-shot adaptation. See [5] for more details. Authors need to discuss this more from a preliminary angle as this is a background in the research literature.

- Figure 4: I understand the authors aim to show the persistency between images of $G_{\mathcal{S}}$ and $G_{\mathcal{T}}$ on the left side of the figure, but adding samples from $G_{\mathcal{S}}$ on the right side could be confusing for some readers. They might think two different source generators are used, even though the samples on the right side also come from the same $G_{\mathcal{S}}$.

$ $

3. **The linear composition of directional vectors is one of the paper's major contributions, but it is not explored properly.**  More specifically, in 3.3, authors aim to discuss the interpolation of the different directions in the CLIP embedding space, and then use this property to combine different target domains. However, instead of having a more systematic analysis or empirical study, the results in Figure 3 are kind of a version of the proposed method.

$ $

4. **Experimental results are not convincing in their current form.** The following issues need to be addressed:

- Using Dino (I assume Dino-V2) for patch-level feature matching between images generated by source and adapted generators makes sense given the capabilities of the Dino in this task. However, **leveraging a powerful model like this makes the comparison with NADA unfair**. For a fair comparison, it is better to have an ablation study and add a similar leverage to NADA in addition to the original NADA implementation. This could also provide good insights for the readers of the paper.

- Qualitative results in Figure 5 are not convincing:
     * how FHDA is suffering from severe mode collapse? usually in few-shot regimes, the mode collapse is observed as generating exactly the same training images which is not the case with FHDA! In fact, it maintains a good amount of diversity in terms of hairstyle, accessories and ....
     * The proposed method has some issues with adapting the style of some of the reference domains. For example, in row 3, the style of NADA seems to be more similar to a cartoonish image rather than the proposed method, or in line 2, the proposed method is not gaining that much style of the HulK or wooden sculpture while NADA at least gains one.
     * In addition, NADA is specifically proposed for the zero-shot setup (using text modality). For a fair one-shot comparison, you should include approaches like MindTheGap [4].

- Quantitative results in Tables 1 and 2 are not also very accurate and convincing for me. Specifically:
     * I am not sure the proposed average CS-I metric is a good representation of this measurement since the fine-grained information is lost. For example, how much of the generated images in the second raw are similar to the reference Hulk image in Figure 5?
     * For a better quantitative comparison, a user study is critical (a more comprehensive one; not like the one we see in Table 6) to compare these approaches based on the human observers' feedback.
     * Better performance of the proposed method can be explained by using Dino-V2 as an additional source of knowledge while other approaches are not using that extra information.

- Generally, I think in some cases NADA is doing a better job in adapting the style of either one or multiple target domains and the proposed method lacks this property.

$ $

I am willing to increase my score if authors address these concerns properly.



$ $

**References:**

- [1] 'Domain Re-Modulation for Few-Shot Generative Domain Adaptation', NeurIPS 2023

- [2] 'Few-shot hybrid domain adaptation of image generators' ICLR 2024

- [3] 'Domain Expansion of Image Generators' CVPR 2023.

- [4] 'Mind the Gap: Domain Gap Control for Single Shot Domain Adaptation for Generative Adversarial Networks' ICLR 2022

- [5] 'A Survey on Generative Modeling with Limited Data, Few Shots and Zero Shot', arXiv 2023

**Questions:**

Please refer to weaknesses.

---

### Note · Authors · 2024-11-15

I have read and agree with the venue's withdrawal policy on behalf of myself and my co-authors.